# Recent Trends in Applying Ortho-Nitrobenzyl Esters for the Design of Photo-Responsive Polymer Networks

**DOI:** 10.3390/ma13122777

**Published:** 2020-06-19

**Authors:** Angelo Romano, Ignazio Roppolo, Elisabeth Rossegger, Sandra Schlögl, Marco Sangermano

**Affiliations:** 1Department of Applied Science and Technology, Politecnico di Torino, Corso Duca degli Abruzzi 24, 10129 Torino, Italy; angelo.romano@polito.it (A.R.); ignazio.roppolo@polito.it (I.R.); 2Polymer Competence Center Leoben GmbH, Roseggerstrasse 12, Leoben 8700, Austria; Elisabeth.Rossegger@pccl.at (E.R.); Sandra.Schloegl@pccl.at (S.S.)

**Keywords:** ortho-nitrobenzyl ester, stimuli responsive polymers, smart coatings, light responsive coatings, photopatterning

## Abstract

Polymers with light-responsive groups have gained increased attention in the design of functional materials, as they allow changes in polymers properties, on demand, and simply by light exposure. For the synthesis of polymers and polymer networks with photolabile properties, the introduction o-nitrobenzyl alcohol (o-NB) derivatives as light-responsive chromophores has become a convenient and powerful route. Although o-NB groups were successfully exploited in numerous applications, this review pays particular attention to the studies in which they were included as photo-responsive moieties in thin polymer films and functional polymer coatings. The review is divided into four different sections according to the chemical structure of the polymer networks: (i) acrylate and methacrylate; (ii) thiol-click; (iii) epoxy; and (iv) polydimethylsiloxane. We conclude with an outlook of the present challenges and future perspectives of the versatile and unique features of o-NB chemistry.

## 1. Introduction

Responsive molecules that undergo changes in structure or properties upon the application of external stimuli represent one of the most ambitious research areas in many scientific fields such as nanomedicine, biology, and functional coatings [1,2,3,4]. Typical external stimuli include both physical stimuli such as temperature, light, or magnetic field and chemical stimuli such as redox, pH value, or biological [5,6,7]. 

Among these stimuli, light has undoubted advantages in terms of spatial and temporal accuracy, the possibility to be activated and deactivated on demand and photo-triggered reactions typically proceed under mild reaction conditions [8,9]. For these reasons, several photo-responsive molecules were synthetized and exploited in numerous areas ranging from organic synthesis to polymer chemistry [8,9,10,11] 

In a further distinction, light-responsive molecules may undergo through reversible or irreversible reactions, depending on the possibility or not for the molecules to recover their initial structures [8,9,12,13,14]. Prominent chromophores undergoing reversible reactions are azobenzene groups (cis-trans isomerization) or coumarin and anthracene derivatives (reversible cycloaddition reactions) [12,14]. 

In the class of irreversible photo-responsive molecules, ortho-nitrobenzyl alcohol derivatives have become one of the most investigated groups in polymer science, for their versatility in being included in many different polymer networks and applications [15,16,17,18,19]. Initially, they were used as photo-protected groups in organic chemistry. A first application in this sense was presented in the work of Barltrop and co-workers in 1966, in which ortho-nitro benzyl ester groups (o-NBE) were employed as photo-protected groups for carboxylic acid [20]. The photo-deprotection under UV light occurs following this reaction mechanism (Figure 1).

Upon UV irradiation (300–365 nm) the excited o-NBE chromophore abstracts a hydrogen from the methylene or methine carbon in γ-position with the formation of aci-nitro tautomers. This is followed by a molecular rearrangement, with the formation of a benzoisoxaline derivative that subsequently cleaves, yielding a carboxylic acid and an o-nitrosobenzaldehyde as primary photoproducts. Secondary photoreactions involve the formation of azobenzene groups by dimerization of the o-nitrosobenzaldehyde [15,16,17,18,19].

Along with carboxylic acids, the idea of employing o-nitrobenzyl alcohol derivatives as photo-protected groups was extended to several other functional groups such as carbamates and peptides [22,23,24,25]. Various strategies were pursued in order to shift the absorption window of the o-NB chromophores to longer wavelengths, to increase the quantum yield and to prevent the formation of photo-dimerized by-products [16,26,27,28,29,30]. These strategies include chemical modification either on the aromatic ring or at the benzyl position of the linker, and photo-sensitization mechanism [31,32]. The possibility to induce the cleavage of the o-NBE with near infrared light via 2 photon absorption mechanism opened an interesting opportunity for deep-tissue biomedical applications [33,34,35]. 

A first adoption of o-nitrobenzyl alcohol moieties as photolabile groups in polymer chemistry is reported in the work of Petropoulos in 1977 [36]. They were initially introduced in polymer networks for the fabrication of UV-sensitive photoresists and progressively adopted as photocleavable crosslinkers for light-triggered drug delivery or as photocleavable junctions in copolymer blocks [37,38,39,40,41,42,43,44,45,46,47,48,49]. Advancing from bulk properties, the photo-sensitive nature of o-NB groups was used to opto-regulate the surface chemistry [50,51,52]. Examples are the fabrication of polyelectrolyte multilayers or the photopatterning of micro and nano arrays for spatially controlled protein immobilization and cell culture [53,54,55,56]. Other interesting applications include photocleavable self-assembled monolayers (SAMs) and optical devices [57,58,59,60].

Although there are numerous studies and applications of o-NB groups reported in literature, this review highlights the studies in which o-NB chromophores are applied as photosensitive moieties in thin polymer films, with a focus on the works made by the authors in the past years. In order to improve the readability and the understanding through the manuscript, the literature was divided into four main categories, according to the polymer matrix, which included (i) methacrylate and acrylates, (ii) thiol-click networks, (iii) epoxy-based networks, and (iv) polydimethylsiloxane. 

We will conclude summarizing selected key-features of ortho-nitrobenzyl alcohol derivatives in polymer chemistry, highlighting present challenges and future perspectives of this expanding research area.

## 2. Photo-Responsive Acrylate and Methacrylate Polymers

Because of their versatile properties and their commercial availability, acrylate and methacrylate monomers have been widely used in the fabrication of polymers since decades [61]. By carefully selecting monomer structure and their functionalities, polymers can be synthesized for a plethora of industrial applications such as industrial coatings, adhesives, inks, and biomaterials [62,63,64,65]. The high reactivity of the carbon-carbon double bonds together with the possibility to be photo-initiated made (meth) acrylic monomers also the most used compounds in new technologies such as light-activated 3D printing technologies [66,67]. 

The possibility to combine acrylate or methacrylate polymer networks with photolabile groups could give further advantages in terms of “smart” properties and in 4D printing strategies [68]. In the last decades, o-NB alcohol derivatives were employed in several applications based on methacrylate and acrylate networks and the first interest in thin polymer films arose for the fabrication of photoresists [40]. 

A photoresist is a light sensitive polymeric compound, whose solubility properties are locally changed through mask-mediated irradiation [69]. In particular, photolithographic techniques are typically applied to inscribe patterns in thin films of the photoresists. In a subsequent step, the pattern is developed with a proper solvent that either dissolves the irradiated (positive tone photoresist) or the non-irradiated area (negative tone photoresist), leaving the desired architecture that could be then used for further processes (metal deposition, epitaxial growth, surface functionalization, etc.) [70].

Different approaches were adopted in literature in order to exploit o-NBE chemistry for the development of photoresists. 

One of the first work in this sense was made by Reichmanis et al. [71] in the early 80s. They introduced o-nitrobenzyl cholate ester in a polymeric matrix of poly-co-(methyl methacrylate-methacrylic acid) in order to inhibit the network’s solubility. Upon UV light irradiation, the o-nitrobenzyl cholate ester was cleaved into soluble species allowing a selective extraction of the irradiated area in an aqueous alkali developer. Results showed higher resolution (1 micron) compared to other photopatterning techniques. The authors also demonstrated the possibility to increase the reaction quantum yield, by proper substitutions on the o-nitro-benzyl chromophore. 

In another work [72], Reinhold Schwalm compared different co-polymers containing o-nitrobenzyl (meth)acrylates as solution inhibitor, in terms of quantum yields (referred to the isomerization reaction of the nitro groups), resist sensitivity, and contrast. In this study the high influence of the network’s glass transition temperature (Tg) on the ortho-nitro reaction was evidenced, that is a consequence of the high free volume that the o-nitrobenzyl group requires for the photoisomerization reaction. In fact, o-nitrobenzyl isomerization proceeds via some intermediates, which require at least rotational motions of the nitro and benzyl moieties. Thus, relatively large amounts of free volume are necessary in order to obtain high quantum yields. This property is also common in other Norrish type II reactions [73]. Consequently, in order to have high cleavage conversion of the nitro groups, the system should be irradiated above its Tg. According to this logic, the author selected appropriate co-polymer films, optimized the reaction parameters, and was able to fabricate micro-resists with a resolution between 0.75 and 1 µm. 

In the late 1980s, particular interest was also focused on the exploitation of this photocleavage reaction in chemical amplification concepts. Willson and co-workers synthetized poly(carbonate)s with *o*-nitrobenzyl pendant groups, which form free hydroxymethyl pendant groups upon UV exposure [74]. Chemically amplified depolymerization of the illuminated polymer was then obtained during a post-baking step in the presence of a photo acid. A route towards positive chemical amplification was pursued by Wilkins et al. who described the synthesis of an *o*-NB ester of cholic acid [75]. The ester acts as photosensitive dissolution inhibitor for copolymers of methyl methacrylate and methyl acrylic acid. Deep UV exposure changed the solubility characteristics of the inhibitor since the cleavage products were highly soluble in alkaline solutions and positive tone patterns with a resolution of 0.5 µm were accomplished. 

These works highlight the advantages of o-NBE links in photoresists compared to conventional ones that employed moieties that are unusable at wavelength shorter than 300 nm, such as novolak-quinone diazine [71,76]. Instead, with nitrobenzyl groups, being active to shorter wavelengths, it is possible to reduce light diffraction effect, thus, increasing the definition and resolution of the photoresist [77]. The high reaction quantum yields of o-NB isomerization reduces also the necessity to use high light energy doses that could degrade the matrix producing poor profiles [72]. 

More recently, o-NBE-based photoresists were successfully adopted for the patterning of organic light-emitting diode (OLED) displays and organic thin-film transistors (OTFTs). An interesting work in this sense was made by Taylor and co-workers in 2009 [78]. They synthetized a photoresist based on a co-polymer that contains monomers with highly fluorinated groups and methacrylic nitrobenzyl ester monomers (Figure 2). After photolysis of the o-nitrobenzyl ester groups, the co-polymer became insoluble in hydrofluoroethers solvents allowing selective network dissolutions. Taking advantages from the photo-switchable solubility properties, the synthetized photoresist was employed as photo-pattern system on PEDOT: PSS films. The results showed a sub-micrometer patterning, and, generally higher performance compared to other chemically amplified techniques for PEDOT: PSS photo-patterning. As a further application, they were able to fabricate a field-effect transistor, in which an organic semiconductor material (pentacene) was patterned on the PEDOT: PSS coating.

Although the good resolutions, the typical dissolution mechanism of the previously reported photoresist may include most of the time harsh developing solvents, which could be particularly insidious for applications in biological field. Different approaches were developed in order to use mild buffer solutions for the photoresist’s developing step [55,56,79,80,81]. The basic idea was the following: being carboxylic acids one of the reaction products of the isomerization reaction of o-NBE (Figure 1), they could be negatively charged after deprotonation at neutral or basic pH. Then, the electrostatic charge repulsions could be conveniently exploited in order to dissolve the UV irradiated area in mild aqueous solution at certain pH values. In order to exploit successfully this polyelectrolyte property in aqueous media, the polymer network should be hydrophilic enough to ensure that water penetrates the matrix. 

The possibility to fabricate photoresist with high resolution and in mild reaction conditions were widely explored for the fabrication of protein microarrays. Different works in this sense were conducted by Doh and co-workers [79]. They synthetized a terpolymer (o-NBEMA-MMA-PEGMA) containing an o-nitrobenzyl methacrylate (o-NBEMA), methyl methacrylate (MMA), and poly(ethylene glycol)methacrylate (PEGMA) for preparing thin photo-responsive films. The patterns were accomplished after UV illumination, by dissolving the irradiated polymer area in a phosphate buffer solution with a pH above 6.6. Further conjugation of biotin to the hydroxyl end groups of PEGMA units allowed selective immobilization of streptavidin on the surface, enabling a multicomponent protein patterning (Figure 3). 

The same group fabricated micropatterns for protein adhesion and cells culture, with microscope projection photolithography [80]. In a first step, they coated a biotinylated substrate (for proteins adhesion), on the top of which they deposited a photo-responsive terpolymer (42 wt% DMNBMA (4,5-dimethoxy-2-nitrobenzyl methacrylate), 24 wt% MMA, and 34 wt% PEGMA) by spin casting. The photoresist design was then inscribed through microscope UV light projection, dissolving the irradiated terpolymer coating. The exposed underlying biotin groups were subsequently exploited for a selective immobilization of streptavidin proteins. By repeating the previous procedure, the authors were able to immobilize different type of proteins in a selective manner, and in micro-arrays with high-resolution capability (1 µm). In a further step they successfully extended the procedure to the fabrication of micro-scaffolds for immune cells (Figure 4a). 

In a further work [81], they extended the concept to the fabrication of a multi-topographical structures, combining two different lithographic techniques: capillary force lithography and microscope projection photolithography (Figure 4b). In a double step procedure, they coated a first pattern of the previously synthetized terpolymer on a biotinylated substrate using capillary force lithography, and they immobilized a first type of streptavidin protein on the uncovered biotinylated area. In a subsequent step, they projected a second pattern with UV light projection, and immobilised a second type of protein on the exposed area. Combining these techniques, they were able to design more complex topographical structures for protein microarrays, which were used as a scaffold for studying cells behavior in a multi-topographical domain. This study reported that for a colon cancer cells culture (SW480) the nanoscale topographical cues, as compared to microscale topographical cues, had an ability to elicit the selective adhesion of the cells.

Because of their polyelectrolytes nature, o-NBE groups were also used for the fabrication of PEMs (polyelectrolyte multilayers) [55,56]. 

PEMs are multi-layered films in which layers of polycations and layers of polyanions are alternated [82,83]. The electrostatic attraction between oppositely charged macromolecules provides the enthalpic driving force for the layer assembly [84]. In a responsive PEM network, an external trigger (such as pH, light or temperature) could selectively change the charge interaction force among the layers, thus promoting a de-assembly mechanism in proper solvents [85]. 

As regard to this concept, Thomas and co-workers employed an o-NBE methacrylate monomer for the fabrication of photo-responsive PEMs networks. In a first work [55], photo-responsive PEMs networks were fabricated overlapping two domains with different wavelength responsiveness (Figure 5). Their idea was to selectively de-assembly the two PEMs domains using light sources operating at different wavelengths. In order to meet this challenge, they included two photolabile groups; a dialkylamino-coumarin ester molecule, which is responsive to visible light, and o-nitrobenzyl molecules, which are excited with UV light. In particular, the first domain alternates layers of poly(styrene sulfonate) (PSS), which contain polyanions ($SO_{3}^{-}$), with layers of 2-(N,N dimethylamino)ethyl methacrylate-o-NBE-methacrylates), which contain polycations. A second domain alternates between PSS and co-2-(N,N dimethylamino)ethyl methacrylate-dialkylaminocoumarin methacrylate, as shown in Figure 5. 

The layers electrostatic interaction was then selectively disrupted by irradiating in series the two domains with two distinct wavelength sources: λ > 400 nm for the photolysis of the coumarin ester and λ < 400 nm for the photocleavage of the o-NBE groups, taking advantage from the photo-release of ionizable carboxylic acid species. The layers de-assembly was promoted in a buffer solution since polyanions are formed by the de-protonation of the carboxylic acid groups in neutral and basic environment. 

By including two different classes of pendant dyes in the two distinct domains, they demonstrated high wavelength selectivity in the de-assembly process through fluorescent measurements.

In a follow-up work [56] the authors increased the level of selectivity by adding two other classes of chromophores. Following the same procedure, four different domains were fabricated containing: dialkylamino coumarin (DEACM) groups (λ < 450 nm), two ortho-nitrobenzyl ester groups (λ < 400 nm), and para-methoxyphenacyl (PMP) groups (λ < 320 nm).

In order to enhance the level of selectivity, they employed two different types of o-nitrobenzyl molecules, which have similar absorbance spectra but different photolysis quantum yields: a 2-nitrobenzyl ester (NBE) unsubstituted at the benzylic position, and an α-methyl-2-nitrobenzyl ester (MNBE). The MNBE has a methyl group at the benzylic position, and a quantum yield of photolysis approximately five times higher than the unsubstituted NBE group. In this way, they were able to design complex patterns by varying both wavelength and intensity of the light source.

The previously mentioned procedures are mainly based on the idea to use o-nitrobenzyl links, in order to spatially control the solubility of thin polymer films in organic solvents or in aqueous solutions. Although good resolutions are achieved, there are some disadvantages, which include the impossibility to inscribe 2.5D or 3D profiles, because of the limitations of the photo-lithography techniques [86]. Recent works have proposed a new original approach, that consists in exploiting a direct cleavage of the o-NBE chromophores and surface ablation, by an UV laser source [87]. In this way, the photoresist development is accomplished with a direct evaporation of the cleaved network from the coating surface, thus avoiding the solvent developing step. 

In a recent and very interesting work, Batchelor et al. [88] adopted this idea combing a 3D printing process with a subsequent network erasing, only changing the wavelength emission of a 2 photons laser source. They synthetized a diacrylate crosslinker with o-NBE groups. In the first step, they 3D printed micrometric structures using a two photons emission laser source (900 nm). In a second step, they selectively erased the structure by tuning the laser wavelength emission (700 nm). In particular, the authors fabricated a cubic structure of length 15 µm using wavelength of 900 nm, and in a second step they inscribed submicrometric tunnels, which crosses inside the 3D structure, with a laser irradiation of 700 nm, owing to the two photons photocleavage of o-NBE groups (Figure 6).

o-Nitrobenzyl chemistry was also widely explored as an efficient tool for tuning polymer surface properties in terms of both physical-chemical properties and morphology. 

Properties such as adhesion, cohesion, optical refraction, surface wettability, or morphology on a micro and nanoscale can be selectively tuned upon UV irradiations, because of the light-responsive nature of nitro-benzyl ester moieties, and their reaction products [50,51,52,59,60,89,90]. An interesting work in this sense is reported by Minkyu Kima and Hoyong Chung in 2017 [91].

They fabricated a photo-responsive bio-inspired adhesive based on a terpolymer poly(MDOPA-co-SBMA-co-NBEDM), composed by a zwitterionic polymer poly(sulfobetaine methacrylate) (pSBMA); a DOPA (3,4-dihydroxyphenylalanine) for the adhesion properties, and a photo-responsive methacrylate-co-2-nitro 1,3-benzenedimethanol dimethacrylate (NBDM). The photocleavage of the o-NBE groups in the synthetized terpolymer was conveniently exploited in order to decrease the crosslinking density, and hence reducing the adhesion properties of the polymer network. The results showed a decrease of the starting adhesion strength from 341 kPa to 223 kPa after 30 min of UV irradiation (38% reduction), and a further decrease to 150 kPa after 3 h of UV irradiation (56%). 

In another brilliant work [92], Xue and co-workers reported a photoinduced strategy for regulating the localized growth of swollen microstructures, by coupling photolysis, photopolymerization, and transesterification mechanism together. The material system consisted of a swellable matrix filled by a swelling solution. The swellable matrix was made by a polymer structure composed of 4-hydroxybutyl acrylate (HBA), for imparting good swelling ability, o-nitrobenzyl acrylate as photolabile linkers, and 1,6-hexanediol diacrylate (HDDA) for the transesterification reaction.

The swelling solution was instead composed by 4-hydroxybutyl acrylate (monomer), 1,6-hexanediol diacrylate (crosslinker), I-819 (photoinitiator), and benzenesulfonic acid (BZSA) as transesterification catalyst. The basic swelling mechanism was the following: upon light exposure, the solution on the irradiated area photopolymerizes, and consuming the monomers, creates a concentration gradient of monomers and crosslinkers (swelling solution) to the irradiated areas. In this way, unreacted species diffuse to the irradiated regions and swells the polymer surface. In addition to this mechanism, the photolysis of the o-NBE groups releases ionizable carboxylic acid groups, which enhance the swelling ability of the irradiated region, by expanding the polymer networks via COO^−^–COO^−^ electrostatic repulsions. The combined effect caused a localized swelling of the irradiated network surface, with the formation of micro-pillars (Figure 7). The presence of o-nitrobenzyl ester groups was crucial for enhancing the swelling process. 

The authors further showed that the dimensions of the pillars could be properly designed and controlled by tuning the light spot dimensions and intensity. Moreover, multi-pillars growing (one above the others) could be realized by irradiating the surfaces of existing pillars in several steps. In addition, the thermal effect generated by the photopolymerization triggers transesterification reactions, which release any polymerization-induced mechanical tension in the dynamic networks, increasing pillars’ mechanical properties. Finally, the authors showed different applications of this dynamic networks, which include surface photo-patterning and healing ability on the millimeter level. 

Although several studies report on the use of controlled radical polymerization (CRP) techniques for the polymerization of o-NBE acrylate and methacrylate monomers, there are still some challenges related to the difficulties to synthetize polymers with high molecular weight and low polydispersity [93,94,95,96]. These difficulties are due to the intrinsic nature of the nitro-aromatic groups that act as inhibitors in the radical polymer growths, particularly at high temperatures. These inhibition effects were also found in living polymerization of other molecules that contain nitro aromatic groups [95,97]. Below, we report some works related to CRP polymerization techniques in o-NBE methacrylate and acrylate systems. 

A first exhaustive study of o-nitrobenzyl methacrylate and acrylate monomers polymerized by different controlled radical mediated polymerization, was made by Schumers and co-workers in 2011 [95]. The study analyzed different types of controlled-radical polymerization (CRP) techniques, for o-nitrobenzyl methacrylate (NBMA) and o-nitrobenzyl acrylate (NBA) monomers. Particularly, they analyzed (i) atom transfer radical polymerization (ATRP), (ii) reversible addition-fragmentation chain transfer polymerization (RAFT), and (iii) nitroxide-mediated polymerization (NMP). Considering RAFT polymerization, the authors could polymerize only methacrylate o-NBE monomers with some degree of control (PDI 1.5) but reaching molar masses up to 11,000 g/mol only after 18 h of reaction. The study showed that RAFT polymerization of NBA was almost completely inhibited. This behavior was ascribed to the higher reactivity of the acrylate radicals, in comparison to the methacrylate ones, and therefore higher tendency to undergo to side reactions with the o-nitrobenzyl group. Regarding nitroxide-mediated polymerization (NMP), no polymerization was observed under the employed reaction conditions. According to this study, ATRP proved to be the best method for methacrylate monomers, achieving polymers with molar masses between 12,100 and 16,400 g/mol and with low-polydispersity indices (Mw/Mn < 1.25). With ATPR, o-nitrobenzyl acrylate with low molar masses and high polydispersity can be polymerized.

In a more recent work Soliman et al. [96] improved these results by single electron transfer–living radical polymerization (SET–LRP). In this study, the authors investigated the influence on the reaction kinetic determined by the presence of ligand, CuBr_2_ (used for preventing the formation of high molecular weight contaminant), and Cu(0) wire length used as a catalyzer for starting the reaction.

In this study, the authors polymerized nitrobenzyl acrylate monomers through SET–LRP method, showing the effect on the polymerization kinetics due by the ligand and CuBr_2_ concentration, as well as Cu(0) wire length used as a catalyzer for starting the reaction.

Optimizing these parameters, they were able to synthetize polymers with a molecular weight up to 28,600 g/mol and a narrow distribution (Mw/Mn < 1.2). 

o-NBE moieties were also employed as photocleavable junctions in copolymer blocks, in order to make micelles, or well-defined nano-porous structure in thin coating applications [26,38,48,98]. Block co-polymers comprise two or more chemically distinct homo-polymer units, linked by covalent bonds [99]. They have attracted much interest in polymer chemistry because, in thin polymer coatings, they are able to self-assembly into arrays of microscopic or nanoscopic domains [100,101]. The self-assembly behavior is a consequence of the intrinsic immiscibility between the two polymer blocks, together with the presence of covalent bonds among the monomer units [102].

Under annealing treatment in vapor solvent, the intrinsic block immiscibility promotes a self-assembling process of the co-polymer networks in well-defined and distinct domains. Photocleavable junctions among the two polymer blocks were successfully exploited as powerful tool, in order to selectively photocleave only one domain of the co-polymer structure, thus realizing porous-ordered domains or spatial-controlled surface post-modifications treatments [93,103]. 

In an interesting work, Schumers and co-workers [93] synthetized a copolymer block (P(NBA-r-AA)-b-PS), based on a photo-responsive poly(2-nitrobenzyl acrylate-random-acrylic acid) unit, and a polystyrene block, PS. After showing a convenient synthesis technique for the block copolymers, they employed the (P(NBA-r-AA)-b-PS) for micelles preparation, by UV irradiating the co-polymer block in a chloroform solution. Because of the UV-induced photo-deprotection mechanism of insoluble acrylic acid functions were released, which promoted co-polymer self-assembling with the formation of micelles with PAA core and a PS corona structure (Figure 8a). TEM characterization showed the formation of 13 nm radius micelles, after 5 h of irradiation at 300 nm and 38.5 mW/cm^2^ of light intensity. The authors were also able to encapsulate, and then photo-release, coumarin 343 as a fluorescent dye. In a further step, thin coatings with a cylindrical morphology were accomplished by self-assembling the P(NBA-r-AA) 74-b-PS581 block copolymers onto silicon substrates (Figure 8b). The authors also studied the influence, in the annealing process, that different solvents had on the co-polymers self-assembling. After UV irradiation, they were able to selectively photo-cleave and extract the o-NBE domains, therefore obtaining well-ordered hollow structures (Figure 8c).

The control of the self-assembly process of the copolymer blocks, and domains orientation, still represents an ambitious challenge in coating applications. Several works have shown the influence of the substrate on the orientation of the structure domains of a coated block co-polymer [104,105]. In a recent work, Sol and co-workers [106] proposed an interesting approach in order to control the domains’ orientation of a coated co-polymer blocks, by photo-controlling the substrate wettability. They fabricated a substrate with photo-responsive wettability based on a block copolymer of poly(styrene-r-2-nitrobenzyl methacrylate-r-glycidyl methacrylate). Because of the release of hydrophilic carboxylic acids under UV light, they were able to change the substrate’s surface polarity. In a subsequent step, by coating block copolymers of poly(styrene-b-methyl methacrylate), they demonstrated that the different substrate wettability was a crucial factor in the self-assemble process of the coated block co-polymer, co-(PS-b-MM), into lamellar or cylinder-forming domains. Particularly, they were able to control the domains’ orientation, from perpendicular to parallel orientation, changing the surface polarity by appropriate UV exposure dose.

In order to enhance the stimuli-responsiveness and the level of selectivity in changing polymer properties, some researches have focused on the design of polymer networks with multi-responsive properties [44,107,108]. 

Adopting a very original approach, Ionov and Diez [109] fabricated a polymer network that combines thermo- and photo-responsive properties by incorporating o-NBE acrylate photocleavable monomers into thermo-responsive poly(N-isopropylacrylamide) (PNIPAM) chains. The photocleavage of the o-NBE groups, upon UV irradiation, was conveniently exploited in order to tune the thermo-responsive behavior of the o-NBE-PNIPAM networks. In particular, the o-NBE photocleavage reaction influenced the low critical solution temperature (LCST), defined as the temperature below which the network is soluble in aqueous solutions. The results showed a decrease of the LCST in aqueous solutions of 50 ℃ after UV irradiation, because of the formation of hydrophilic carboxylic acid species. This temperature-responsive behavior was used in order to sequentially remove irradiated and non-irradiated areas of a photo-patterned surface by carrying out developing steps at different temperatures. In a subsequent step, they were able to selectively immobilize proteins on the patterned surfaces.

## 3. Photo-Responsive “Thiol-Tlick” Networks

Introduced in 2001 by Kolb and co-workers, click chemistry refers to a group of reactions in organic chemistry, that are characterized by fast reaction rates, regioselectivity and stereospecificity, high yields and mild reaction conditions [110]. The term “thiol-click” refers to a large subcategory of click reactions, in which monomers with thiol functionalities react with different type of co-monomers via radical or catalyzed reactions [111]. The reason why they have attracted particular interest in polymer chemistry, is related to their fast reaction rate and the high final monomers conversion. In addition, thiol-click reactions do not suffer from oxygen or water inhibition [111,112,113,114]. The step-growth polymerization mechanism further leads to homogeneous polymer matrix with low shrinkage stresses and narrow glass transition regions [111,112,113,114,115,116]. 

Taking advantage of the unique properties of thiol-click reactions and the versatile nature of o-NBE chemistry, our research group designed various photocleavable “thiol-click” networks with photosensitive o-nitrobenzyl ester moieties [21,87,89,90,117]. For this purpose, a series of o-nitrobenzyl ester monomers with terminal functionalities such as alkene, acrylate, epoxy, and alkyne were synthetized. Thiol-click networks were then prepared by light-induced reaction with different thiol crosslinker (Figure 9).

The design of photocleavable “thiol-click” photopolymers by introducing o-NBE chromophores was reported by Radl and co-workers in 2017 [117]. Radical-mediated thiol-ene photopolymerization of a photo-sensitive alkene(2-nitro-1,4-phenylene)bis(methylene) acetate with multi-functional thiols yielded polymer networks with photosensitive covalent links. The curing of the monomers was carried out by visible light exposure using phenyl bis(2,4,6-trimethylbenzoyl)-phosphine oxide as long wavelength absorbing photoinitiator (λ = 360–440 nm). The absorbance of the initiating system did not interfere with the absorbance of the *o*-NBE links, as vinyl-NBE absorbs below 380 nm. Thus, efficient photoinitiation of the thiol-ene reaction was accomplished without electronically exciting the photosensitive vinyl-NBE. Going beyond the fabrication of positive tone patterns, the spatial and temporal control of the photocuring process enabled the fabrication of complex two-dimensional polymer patterns with photo-responsive properties. Microstructures with a resolution of 4 µm were accomplished by dissolving the cleaved networks in organic solvents (Figure 10).

Inspired by this work, Romano and co-workers [21] synthetized a photo-responsive thiol-epoxy network by using a photolabile o-NBE derivative with terminal epoxy groups (epoxy-NBE) (Figure 9). Crosslinking of epoxy-NBE with multi-functional thiols was accomplished through a photo-base catalyzed nucleophilic ring opening of the epoxy monomers. The chosen catalyst was a photo-latent base, which releases strong amidine-type base (pH around 13) under light exposure, thus initiating the thiol-epoxy addition reaction. The spectral sensitivity of the photo-latent base was extended to the visible light region by adding a selected photosensitizer to the resin formulation. With this approach the initiation of the crosslinking reaction was carried out with light in the visible wavelength region without interfering with the absorbance of the o-NBE groups. In a second step, the o-NBE groups selectively cleaved upon UV exposure, obtaining a well-defined network degradation. Positive tone photo-patterns with a good resolution were inscribed into thin films of the thiol-epoxy networks by lithography techniques. Going a step beyond, the photo-release of hydrophilic carboxylic acid species was exploited in order to tune the surface wettability. Contact angle measurements revealed a change of the water contact angle from 75° to 62° if the UV irradiation was carried out in nitrogen atmosphere. However, nearly fully wettable surfaces were obtained with water contact angles lower than 10° if the UV exposure was performed under air. The pronounced shift of the water contact angle under air was mainly attributed to a photo-oxidation of the polymer surface.

Rossegger and co-workers [89] exploited the significant change in surface wettability to photo-tune the surface properties of thiol-click networks. In particular, they demonstrated the possibility to drive water droplets across the surfaces through a photo-inscribed wettability gradient. They synthetized o-nitrobenzyl alcohol derivatives with terminal alkyne groups, which were then crosslinked across multi-functional thiols, by a photo-induced thiol-yne reaction. In order to maintain orthogonality between the crosslinking and the cleavage step, the curing reaction was initiated by visible light exposure, using phenyl bis(2,4,6-trimethylbenzoyl)-phosphine oxide as photoinitiator. They showed that the surface wettability can be controlled over a broad range, with water contact angles ranging from 97° to 27° upon UV irradiation under air, and from 97° to 82° upon UV irradiation in nitrogen atmosphere. The authors demonstrated with XPS measurements that the higher wettability change in air relies on the photooxidation of the unreacted thiol groups yielding sulfonic acid moieties. The high wettability gradient generated upon gradual UV exposure together with a Laplace pressure gradient were used to move water droplets on the sample’s surface. For the preparation of this multi-gradient surfaces, a V-shaped pattern geometry with a lengthwise wettability gradient was inscribed into the thin films by photolithography (Figure 11). Because of this design, they were able to move a 2 µL water droplet over a distance of 10 mm; not only on a planar polymer surface but also on surface, which were tilted (20°) or turned upside down.

The wettability of solid surfaces mainly depends on the chemical surface composition and on the microstructure morphology [118]. It is well-known, through the Wenzel equations and other physical models, that surface roughness influences the wettability properties compared to an ideal planar surface with the same chemical composition [118,119]. In particular, Picraux et al. reported that an appropriate surface roughness amplifies the light-induced change in water contact angle of a photo-responsive silica surface [120].

Taking advantages of this effect, in a further work, Rossegger and co-workers [90] increased the wettability gradient by introducing a needle-like micropatterns on the sample surface, with visible light-assisted nanoimprint lithography (NIL). In order to increase the mechanical properties of the coating, for the light-assisted NIL procedure, they prepared a thiol-acrylate photopolymer with a higher Tg. For the sample preparation, an o-nitrobenzyl alcohol derivative with terminal acrylate groups (acrylate-NBE) was synthetized and cured upon visible light exposure in the presence of multi-functional thiols, isobornyl acrylate, and a fluorinated methacrylate. The combined effect of the needle-like microstructures and the fluorine groups increased the surface hydrophobicity, with static water contact angle of 140°. Under UV light irradiation, not only the chemical surface composition but also a photoablation of the microstructures was observed (Figure 12). With this combined effect, they were able to increase the surface hydrophilicity in a controlled manner and obtained photopolymer surfaces with a water contact angle of 7°. The higher wettability gradient, in combination with a Laplace pressure gradient (V-shaped pattern), was exploited in order to move a water droplet over a distance of 22 mm, double than the distance reported in the previous work (Figure 12).

Some general considerations could be made concerning the o-NBE cleavage reaction in the analyzed thiol-click networks. In all previously cited works, the o-NBE cleavage reaction proceeds with fast rate and high reaction yields. This result could be mainly attributed to the typical low Tg, and thus high network flexibility, of the thiol-click networks, which is an effect of the thioether bonds. As already reported in the work of Schwalm, high network mobility allows large free volume for the photoisomerization of the o-NBE groups, thus ensuring high reaction yields [72]. 

Considering instead the formation of soluble species as a function of UV irradiation dose, the reaction proceeds with hyperbolic trend in which, a first initial fast reaction rate is followed by a progressively decrease at higher energy dose (Figure 13a). This result, that is common also in other polymer coatings with photolabile o-NBE groups, is due to the formation of secondary photoproducts [21]. On the one side, they act as inner UV filter and reduce the intensity of the incident light. On the other side, the formation of secondary photoproducts can lead to a re-crosslinking of the polymer chains at prolonged UV exposure [117]. 

This behavior was confirmed by sol-gel analysis of thiol-click networks with o-NBE chromophores as function of UV irradiations dose [21,89,90,117]. As shown in Figure 13b, after a first decrease of the gel content, the network followed a re-crosslinking process, thus decreasing the amount of the extractable soluble species. As shown by solid state NMR experiments, this mechanism is mainly related to azobenzene side products, which are formed by a dimerization of the nitroso-benzaldehyde. These side reactions limit the performance in photoresist applications.

In order to overcome these limitations, Romano and co-workers [87] recently proposed a new patterning approach based on direct laser writing with a UV-laser. For this purpose, they prepared photocleavable thiol-ene networks by photo-curing a trifunctional thiol with vinyl-NBE (Figure 9). In a further step, the network degradation was selectively accomplished through a laser beam source with a wavelength of 375 nm (Figure 14). Contrary to conventional photolithography, in which 2D patterns are inscribed by photomasking techniques and the soluble species are extracted by means of solvents in the development step, in this work a 2.5D profile was simply manufactured by modulating the energy dose of the laser beam in dry conditions, by direct surface ablation of the cleaved network. The direct laser writing offered a further advantage in terms of secondary reaction products. As previously mentioned, prolonged UV exposure significantly limits the amount of extractable soluble species because of the formation of side products and re-crosslinking of the networks. In contrast, with laser irradiation technique, the cleaved off polymer chains are immediately removed by evaporation during the laser writing process, without the possibility for the degraded network to go through secondary photoreactions. In a further step, the authors showed the possibility to tailor the presence of carboxylic acids on the network surface by tuning the laser energy dose. The carboxylic acid groups were further employed as reactor anchor points for immobilizing a conjugated protein (Alexa-546). The results showed a selective anchoring of the fluorescent proteins on sample surface only on the irradiated areas, and within a density correlating to the laser energy irradiation dose and thus, the number of carboxylic acid groups present on the surface. 

Thiol-click reactions have been also widely explored as facile and efficient coupling procedures in surface modifications [121]. As regard to this, Li and co-workers [122] accomplished photolabile network containing both o-NBE and “clickable” groups, in order to realize photoresists with “click” modifiable surfaces (Figure 15). They synthetized three different monomers containing o-NBE links and with three different functionalities; allyl (M1), propargyl (M2), and epoxy (M3). In a second step, polymer chains were accomplished through Passerini MCP reactions, containing the synthetized monomers (M1, M2, and M3), 1,6-hexanedioic acid, and 1,6-diisocyanohexane. The synthetized polymer chains were then crosslinked through amine-epoxy reactions, coupling the epoxy groups (M3) of the polymer chains and diethylenetriamine monomers. The o-NBE groups of the amine-epoxy networks were then exploited as decrosslinking points for fabricating positive tone photoresists. In a further step, the allyl and propargyl groups on the photoresist surface were used as “clickable” points, for anchoring fluorescent molecules with copper(I)-catalyzed azide alkyne cycloaddition (CuAAC), and thiol−ene reactions (Figure 15).

## 4. Photo-Responsive Epoxy Networks 

Although there are several studies reporting the use of *o*-nitrobenzyl ester chromophores in acrylate and methacrylate coatings, there are few works in literature on their use in epoxy-based networks [123,124,125].

Because of their salient properties, such as low shrinkage upon polymerization, insensitivity to oxygen inhibition, good mechanical strength and adhesions to various substrates, epoxy resins have become popular materials in numerous fields of applications ranging from lightweight composites, functional coatings, electronics, and adhesives [126]. The curing of epoxy monomers either proceeds in the presence of hardeners (e.g., amines, amides, anhydrides, phenols, thiols, and imidazoles) or through anionic or cationic homo-polymerization [127]. In the last years, cationic curing of epoxy resins has gained increased attention since the initiator are usually inert in the presence of epoxy monomers, and could be activated on demand by external stimuli such as temperature or light [127]. In particular, the remarkable work of James Crivello on diarylsulfonium and triarylsulfonium salts, which can be used as cationic photoinitiators for the homo-polymerization of epoxy monomers, has become the starting point for studying photo-triggered curing reactions of epoxy networks [126,128].

A first work on photo-responsive epoxy networks containing o-NBE moieties was made by Radl and co-workers in 2015 [124]. In this study, a di-functional epoxy monomer containing o-NBE groups (epoxy-NBE) was synthesized and thermally cured with an anhydride hardener. In a further step, positive tone photoresists with pattern resolution in the range of 8 µm were inscribed, taking advantages of the photo-cleavage of o-NBE groups. In order to study the applicability also in recycling concepts, the o-NBE photo-cleavage reaction was also exploited in order to promote network degradability in thicker coatings (1 mm), and for reducing the matrix-fiber adhesions in polymer-based composites. Mechanical tests showed a good degradation behavior of the UV irradiated coatings, with a depletion of the storage modulus and glass transition temperature with increasing UV exposure dose (see Figure 16). Additionally, single fiber pull-out tests revealed a significant decrease of the interfacial adhesion at the fiber-matrix interface due to the photo-triggered cleavage reaction.

In a follow-up study, the same working group [125] highlighted the possibility to cure epoxy-NBE through a cationic ring opening mechanism using an appropriate photoacid generator as initiator. Two different approaches were explored to activate the photoacid generator without premature cleavage of the o-NBE links, which absorb between 300 and 365 nm. The first approach included the use of a radical promoted sensitization mechanism of the photoacid generator, enabling an excitation upon visible light exposure (λ > 400 nm). The second approach involved a photo-initiation mechanism by sulfonic acids, formed by a direct photolysis of *N*-hydroxynaphthalimide triflate upon deep UV exposure. In both concepts, the cationic ring opening reaction progressed very slowly and a thermal post-treatment was necessary in order to obtain high final monomer conversions. These results could be related to the low reactivity of the epoxy monomers because of the presence of the o-NBE groups. A possible explanation can be found in a previous work of Crivello and Sangermano [129], which demonstrated that the presence of nucleophilic ester groups decreases the reactivity of epoxy monomers in cationic ring opening reactions. However, after thermal post-treatment, epoxy networks with stimuli-responsive properties were successfully obtained, and positive tone photoresists with pattern definition of 20 µm were inscribed onto the polymer surface. 

Regarding the photo-cleavage reaction of the o-NBE groups in epoxy networks, the trends and kinetics are comparable to the previously cited thiol-click networks. In order to better understand how the photocleavage reaction is influenced by the network properties, Giebler and co-workers [123] recently studied different epoxy-anhydride networks containing o-NBE groups. In particular, they thermally cured epoxy-NBE with different types of cycloaliphatic anhydrides in order to obtain networks with varying thermo-mechanical properties. Their results demonstrated that the networks with glass transition temperature (Tg) below room temperature exhibited a higher cleavage reaction rate, but were more prone to undergo secondary photoreactions, and therefore, leading to a re-crosslinking of the cleaved polymer chains. This result is easily explained considering that, higher network flexibility allows large free volume space for the photolysis, but at the same time, higher probability for the photocleaved chains to couple and to re-crosslink. At the same time, networks with a Tg above room temperature showed lower cleavage reaction rate, but at the same time lower tendency to go through secondary photoreactions. Networks with a higher Tg also exhibited superior resist performance, with a contrast of 1.17 and a resolution of 8 µm. These results are in good agreement with other studies cited in this review and evidence how the polymer network properties are crucial for the photo-cleavage kinetics of the o-NBE groups and in general for the photoresist performance. 

## 5. Photo-Responsive PDMS Networks

Polydimethylsiloxane (PDMS), also known as dimethylpolysiloxane or dimethicone, is by far the most widely used siloxane (“silicone”) elastomer in technical application [130]. Its fame in polymer chemistry stems from the special rheological properties and other remarkable material properties such as nontoxicity, biocompatibility, blood compatibility, elasticity, transparency, and durability [130,131,132]. Because of these unique properties, PDMS-based polymer networks are widely used in numerous applications ranging from electrical insulations to contact lenses. PDMS is also often employed in soft lithography and in 3D printing, for the fabrication of micro-scaffold and microfluidic devices [133]. Despite the widespread use of PDMS, there are almost no studies on the design of photo-responsive PDMS coatings with photolabile o-NBE groups. 

The first study on PDMS polymer networks with o-NBE junctions was conducted by Giebler and co-workers [134] in 2018. In this work, polydimethylsiloxane (PDMS) oligomers with terminal anhydride groups were thermally crosslinked with epoxy-NBE. The introduction of o-NBE groups enabled the introduction of a dual-responsive behavior in the PDMS network, which were degraded either upon UV light exposure or in alkaline environment. In addition to the well-known UV degradation mechanism, the hydrolysis-sensitive ester groups of the epoxy-NBE crosslinker undergo hydrolytic degradation in alkaline environments. As regard to this, the results demonstrated a complete hydrolytic network degradation within few hours or some days, as a function of the NaOH molar content in the alkaline solution. In order to show the dual-responsiveness, the o-NBE groups were also employed for a selective photo-degradation, and positive tone photoresist with a resolution of 50 µm were inscribed into the PDMS surface. However, re-crosslinking reactions because of the side reactions were observed at high UV energy dose. Taking advantages from the reformed crosslinks that showed less sensitivity to hydrolytic degradation, the authors were able to inscribe also negative tone photoresist microstructures, in which the not irradiated area were dissolved in alkaline environment (Figure 17). 

## 6. Conclusions

Because of their versatility, o-nitrobenzyl derivatives were successfully exploited in numerous polymer applications and various studies have shown the potential of o-NB chemistry in the design of different photo-responsive polymer networks. In this review, we highlighted the latest researches on o-nitrobenzyl ester chemistry with a focus on thin polymer films applications. The first application in polymer coatings concerned the design of photoresists with high contrast and resolution. The possibility to develop the photoresists in aqueous solvents arose the interest also in biological fields such as protein and cell microarrays. The photo-release of protected groups, such as carboxylic acid, was exploited for opto-regulating the surface’s chemistry in different applications such as SAMs or surface tunable wettability. o-NB groups were also successfully employed in “click” polymer network, hence combining all the advantages of click chemistry with the light responsiveness of o-nitrobenzyl groups. In a recent work, we showed the potentiality in using UV-laser source in order to develop photoresist with 2.5D patterns profiles and in dry conditions, avoiding the use of harsh solvents in the developing step. The possibility to induce the photocleavage through two photons laser sources also opened interesting opportunities in nano-engineering applications which still need to be discovered. Despite the all cited recent progress, present challenges concern the difficulties to induce the photo-cleavage of the nitro groups in thicker coatings due to the presence of byproducts that act as inner filter, and generally, because of the low penetration of UV light. Recent works partially overcame this limitation by two photons emission source or with the use of photobleaching compounds, but there are still some challenges related to the utilization in tissue engineering applications which are mainly due to usual toxicity of these compounds or to the low quantum yields efficiency. 

An interesting possible solution in this sense was proposed by Pirrung and co-workers that proposed a way of dealing with the nitroso side-product, that were trapped in situ by a Diels–Alder reaction with a diene function included at the benzylic site [135].

Other challenges are related to the low polymerization kinetics of the nitro-benzyl monomers which limits the applicability in radical controlled polymerization and in 3D printing. RAFT method showed the best results as well as single electron transfer–living radical polymerization (SET–LRP). Other strategies to enhance the polymerization kinetics exploit different o-NB structures as well as different molecular substitutions. We experienced for example how thiol-acrylateNBE monomers compared to thiol-vynilNBE monomers showed better performance in terms of curing kinetic. In this sense, trying different o-NBE monomers functionalities or also polymerization technique could partially overcome the low reactivity given by the o-nitrobenzyl moieties. Overcoming these listed limitations will provide new opportunities to apply o-nitrobenzyl chemistry in other different areas of polymer science for the next years. 

## Figures and Tables

**Figure 1 materials-13-02777-f001:**
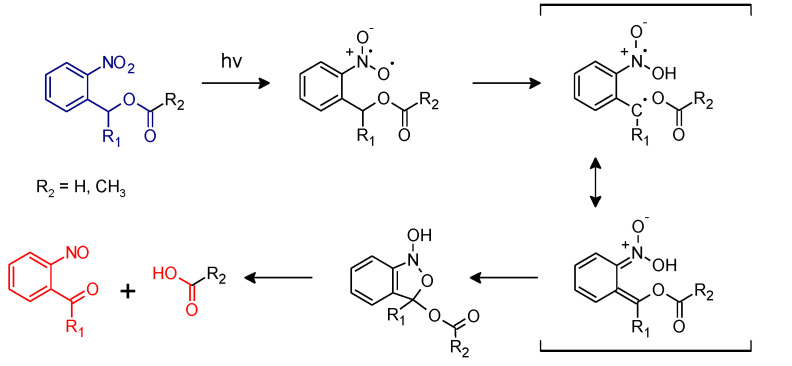
Photoreactions of ortho-nitro benzyl esters (Reproduced by [21] published by the Royal Society of Chemistry).

**Figure 2 materials-13-02777-f002:**
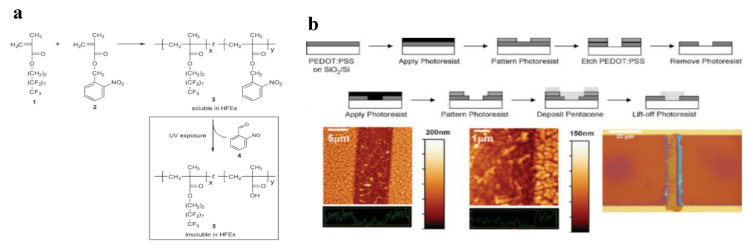
(**a**) Synthesis of the UV-sensitive acid-stable polymer resist 3 and photoinduced deprotection reaction to polymeric carboxylic acid; (**b**) schematic device fabrication of PEDOT: PSS/pentacene bottom-contact OTFT; and optical image of the OTFT (Reprinted with the permission from Reference [78]).

**Figure 3 materials-13-02777-f003:**
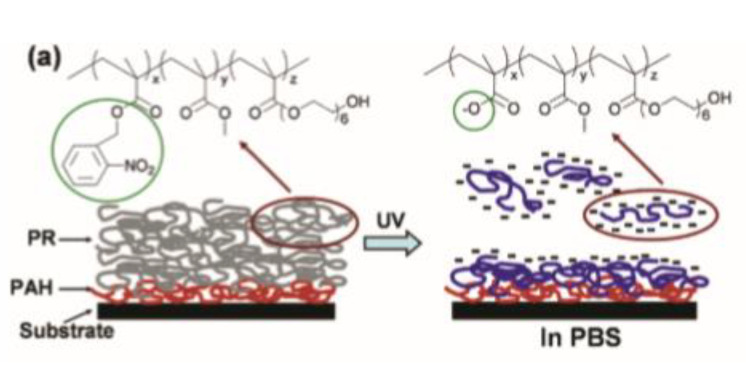
Chemical structure of Photoresist (PR) (**a**) spin-coated on Poly(allylamine) hydrochloride(PAH) and its mechanism for in situ polyelectrolyte bilayer formation (Reprinted with the permission from reference [79] Copyright 2004 American Chemical Society).

**Figure 4 materials-13-02777-f004:**
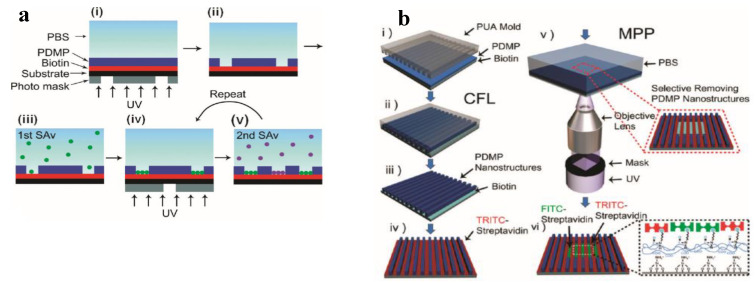
(**a**) Schematic of multiple streptavidin (SAv) patterning (Reprinted with the permission from reference [80] Copyright 2010 American Chemical Society). (**b**) Schematic diagram of the sequential fabrication of multiscale, multicomponent protein-patterned surfaces by combining CFL and MPP (Reprinted with the permission from reference [81] Copyright 2011 American Chemical Society).

**Figure 5 materials-13-02777-f005:**
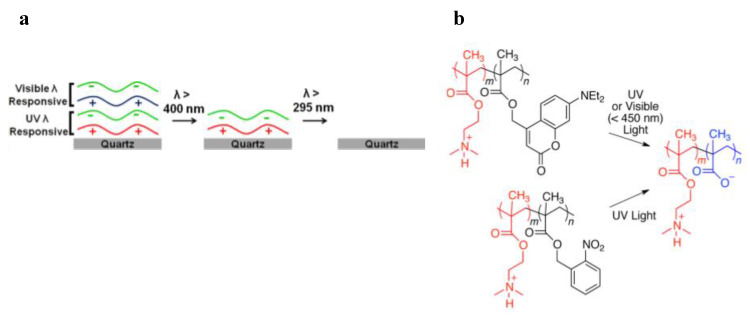
(**a**) Schematic illustration of the de-assembly process through different wavelength irradiations. (**b**) Design of photosensitive polycations that reduced net charge upon irradiation with either visible (coumarinyl groups) or UV (coumarinyl or nitrobenzyl groups) light (Reprinted with the permission from reference [55] Copyright 2014 American Chemical Society).

**Figure 6 materials-13-02777-f006:**
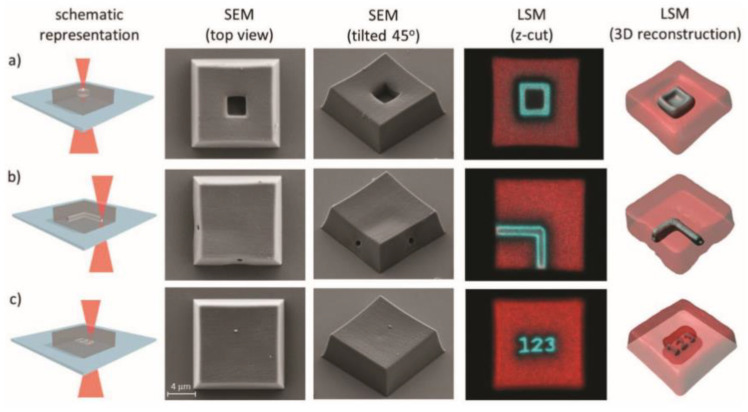
Laser erasing schematic procedure and SEM pictures of the printed micrometric (15 × 15 × 5 µm^3^) blocks using a 700 nm femtosecond laser. (**a**) 3 ×3 × 3 µm^3^ block, (**b**) 90° 1 µm diameter tunnel, and (**c**) simple text “123” (Reprinted with the permission from reference [88]).

**Figure 7 materials-13-02777-f007:**
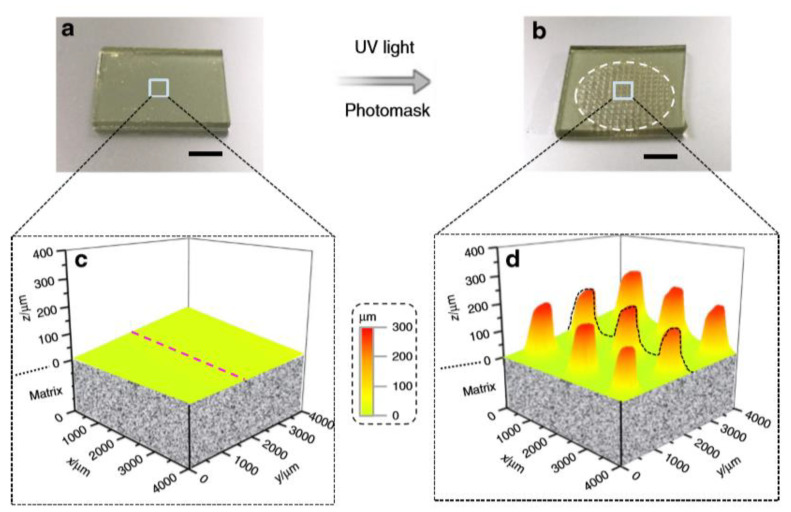
Microstructure pattern grown, (**a**) before UV irradiation, (**b**) after UV irradiation, (**c**) 3D profiles before UV irradiation, (**d**) swollen profile after UV irradiation (Reprinted from reference [92]).

**Figure 8 materials-13-02777-f008:**
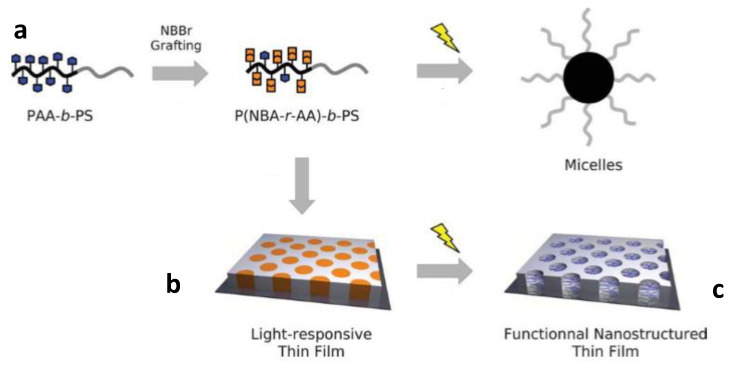
Schematic representation of light-responsive behavior of P(NBA-r-AA)-b-PS photocleavable block copolymers (**a**) light-induced micellization in a selective solvent of PS, (**b**) self-assembly in thin film with a cylindrical morphology, and (**c**) light exposure leading to functional and nanostructured thin films (Reprinted with permission from reference [93]).

**Figure 9 materials-13-02777-f009:**
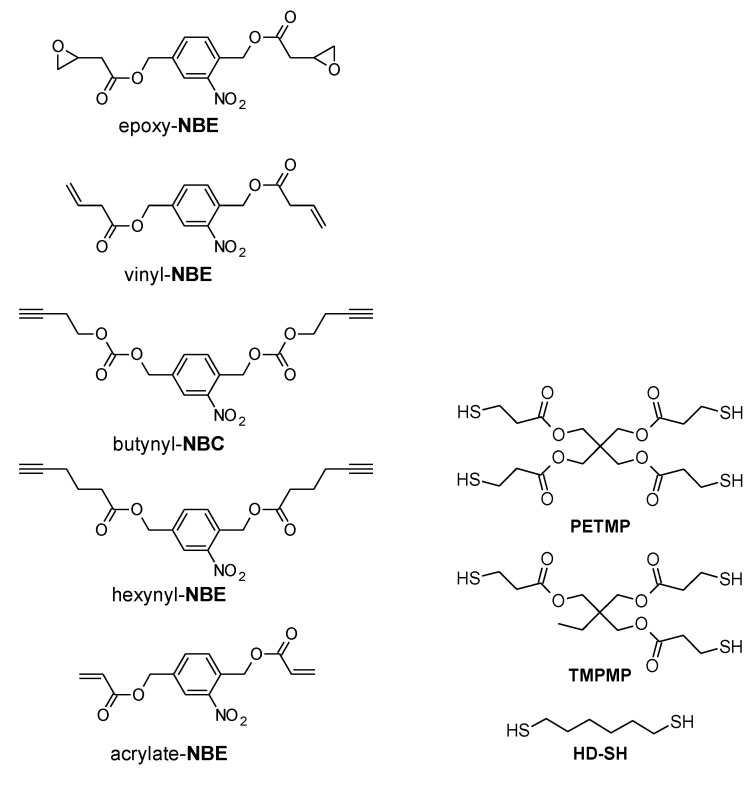
Monomers used for the preparation of the different photo-responsive thiol-click formulations.

**Figure 10 materials-13-02777-f010:**
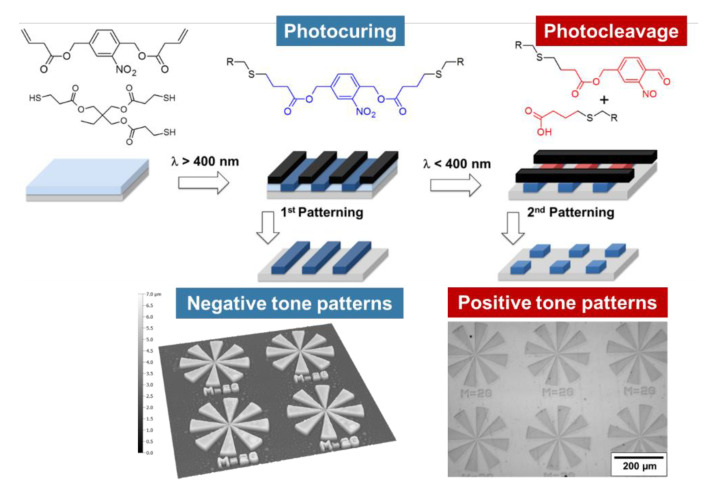
Photo-induced formation and light triggered cleavage of thiol-ene networks for the design of switchable polymer patterns. (Adapted from reference [117] published by the Royal Society of Chemistry).

**Figure 11 materials-13-02777-f011:**
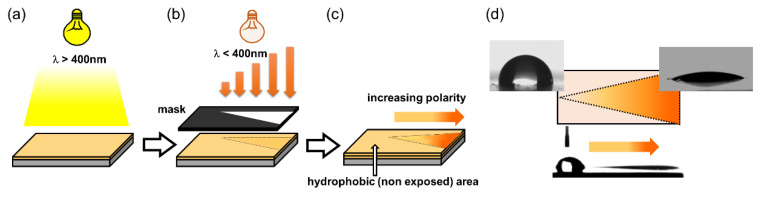
Schematic illustration of (**a**) photopolymerization of the thiol-yne network at long wavelength. (**b**) Patterned and asymmetrical UV exposure to realize (**c**) a wedge-shaped surface area with a lengthwise wettability gradient, which is surrounded by the hydrophobic (not exposed and thus, not-cleaved) thiol-yne network. (**d**) Water contact angle and droplet movement on the irradiated surface (Adapted from reference [89] published by the Royal Society of Chemistry).

**Figure 12 materials-13-02777-f012:**
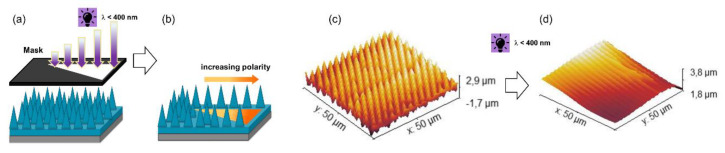
Schematic illustration of (**a,b**) Patterned and asymmetrical UV exposure on NIL thiol-acrylate structure. (**c**) SEM micrographs of the topography of needle-like micropatterns inscribed by visible light assisted NIL (**d**) Surface structures of the photocured photopolymer after 1500 s of UV exposure (269 mW/cm^2^) under air (Adapted from reference [90] published by the Royal Society of Chemistry).

**Figure 13 materials-13-02777-f013:**
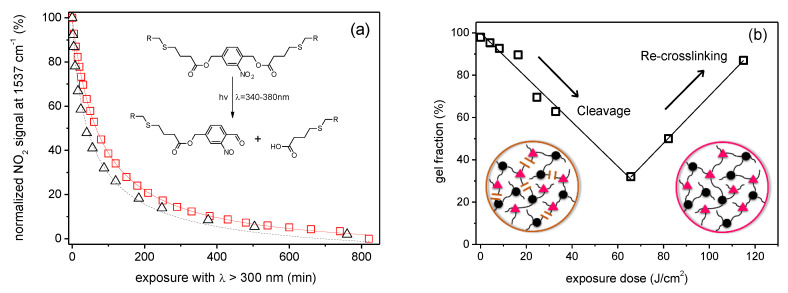
(**a**) Typical ortho-nitrobenzyl cleavage kinetic in thiol-click network. (**b**) Thiol-ene network’s normalized gel fraction after o-NBE photo-cleavage (Adapted from reference [117] published by the Royal Society of Chemistry).

**Figure 14 materials-13-02777-f014:**
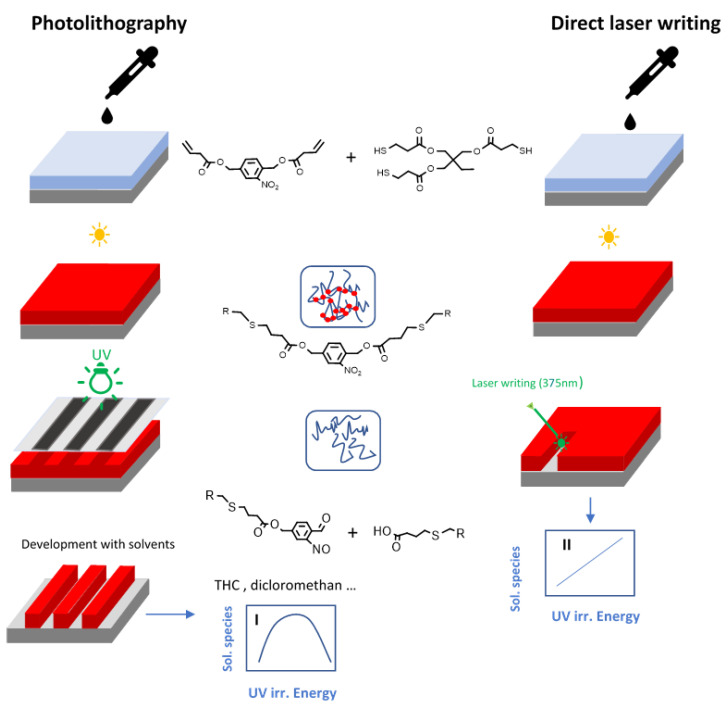
Comparison between photolithography and direct laser writing for the introduction of micropatterns in photosensitive thiol-ene networks (Reprinted with permission from reference [87]).

**Figure 15 materials-13-02777-f015:**
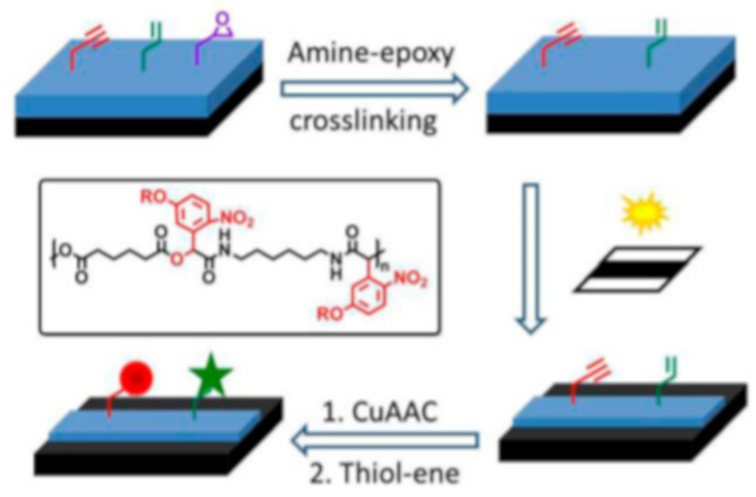
Schematic representation of the process for generation of multifunctional pattern on silicon wafer (Reprinted with the permission from reference [122] Copyright 2014 American Chemical Society).

**Figure 16 materials-13-02777-f016:**
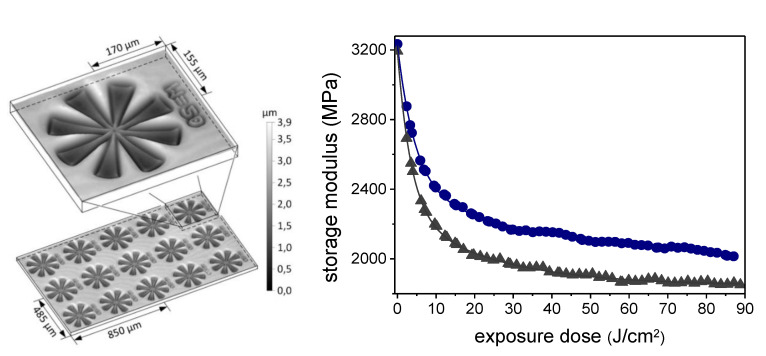
Positive-type photoresists and storage modulus variation of epoxy-NBE network upon UV exposure (Reprinted from reference [124] with the permission of Elsevier).

**Figure 17 materials-13-02777-f017:**
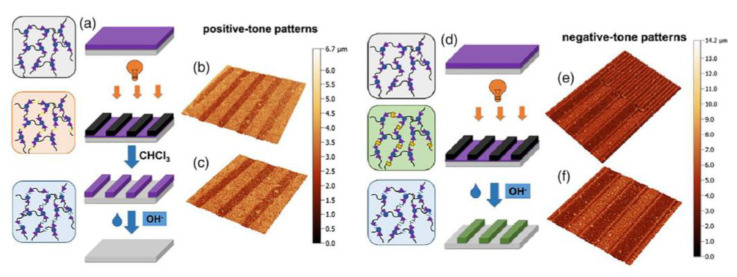
Schematic representation of the formation of a positive-tone resist and (**a**) subsequent removal of the resist by hydrolytic degradation of the PDMS network. (**b** and **c**) Confocal micrographs of positive-tone relief structures (100 μm lines and spaces) inscribed into PDMS-1 by photolithography after the development in chloroform. (**d**) Schematic representation of the formation of a negative-tone resist. (**e** and **f**) Confocal micrographs of negative-tone relief structures (50 and 100 μm lines and spaces) inscribed into polymer network by photolithography after the development in 1 M aqueous NaOH (Reprinted with permission from reference [134]).

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
