# Peer review of "Recent Trends in Applying Ortho-Nitrobenzyl Esters for the Design of Photo-Responsive Polymer Networks"

_materials, 2020, doi:10.3390/ma13122777_

Round 1

Reviewer 1 Report

This paper is a review about o-NBE, especially their application of polymers containing this kind of light sensitive group on various surface scicence, including photoresist and 3D structures. There are a lot of papers related to o-NBE, which is one of the hot spots in the past ten years. Therefore, different types of o-NBE molecular structures are designed to match the needs of different excitation wavelengths and photosensitivity. It is a pity that this paper mainly introduces the simplest o-NBE structure, which is not enough and affects the reading. In this aspect, more references should be provided in the introduction part.

Author Response

Please find the answer in the attached file

Reviewer 2 Report

In this work, a recent review about the studies in which the o-nitrobenzyl alcohol derivatives are used as photo-responsive moieties in thin polymer films and functional polymer coatings is carried out. The manuscript is divided into four categories as function as the polymer matrix. After reviewing the manuscript, the reviewer recommends some minor revisions for that the manuscript is suitable for publication in Materials. The comments and suggestions are included in the attached file.

Author Response

(The authors gave the same response as above.)

Reviewer 3 Report

The authors report an extensive review on studies in which ortho-nitrobenzyl (o-NB) chromophores are applied as photosensitive moieties in thin polymer films. This well-written review summarized different synthesis strategies for polymer networks incorporated with o-NB derivatives as light-responsive chromophores and their applications. However, several articles have previously reviewed the numerous studies and applications of o-NB groups [1,2]. What seems to set this review article apart from the others, was the focus on the progress made by the authors in the last years, reporting the use of o-NB ester chromophores in thiol-click networks, epoxy-based networks and polydimethylsiloxane. This review article will bring a great interest from material science community. I recommend that authors to address the following issues:

1) In the Introduction, the authors overview the history and developments in o-NB derivatives. Can the authors provide some comments on advantages and disadvantages of o-NB compared to other photo-responsive molecules? For instance, photolysis of nitrobenzyl and nitrophenethyl compounds, and their derivatives can form potentially toxic and strongly absorbing by-products, such as o-nitrosobenzaldehyde, and a number of alternative photo-responsive molecules have been developed that don’t suffer from the disadvantages encountered with NB compounds [3].

2) I would suggest organizing the different sections of the manuscript in terms of different functionalities/applications (self-assembled monolayers (SAMs), surface tunable wettability, protein and cell microarrays, etc.) instead of different polymer matrix categories ((i) methacrylate and acrylates, (ii) thiol-click networks, (iii) epoxy-based networks and (iv) polydimethylsiloxane).

3) In the Conclusion section, can the authors summarize/provide some possible solutions to “the low polymerization kinetics of the nitro-benzyl monomers” limitation?

4) Throughout the manuscript several paragraphs relating to the same point are separated, which makes the readability difficult. For instance, on Page 9: “… by Schumers and co-workers in 2011[95]. The study analysed differenttypes…”
5) Please give descriptions for different images (a)-(d) of Figure 7 on page 9.

6) On page 8 lines 272-275 – “They fabricated a photo-responsive bio-inspired adhesive based on a terpolymer poly(MDOPA-272 co-SBMA-co-NBEDM), composed by a zwitterionic polymer poly(sulfobetaine methacrylate) (pSBMA); a DOPA (3,4-dihydroxyphenylalanine) for the adhesion properties, and a photo-responsive methacrylate-co-2-nitro 1,3-benzenedimethanol dimethacrylate (NBDM)”. Please amend the Italics font to have a consistent method.

7) There are some minor typos in the manuscript, mainly around in-text references. For instance, on page 4 line 141.

References

[1]        S. W. Thomas III, “New Applications of Photolabile Nitrobenzyl Groups in Polymers,” Macromol. Chem. Phys., vol. 213, no. 23, pp. 2443–2449, Dec. 2012, doi: 10.1002/macp.201200486.

[2]        A. Abdollahi, H. Roghani-Mamaqani, B. Razavi, and M. Salami-Kalajahi, “The light-controlling of temperature-responsivity in stimuli-responsive polymers,” Polym. Chem., vol. 10, no. 42, pp. 5686–5720, 2019, doi: 10.1039/C9PY00890J.

Author Response

(The authors gave the same response as above.)
